# The Yeast and Hypha Phases of *Candida krusei* Induce the Apoptosis of Bovine Mammary Epithelial Cells via Distinct Signaling Pathways

**DOI:** 10.3390/ani13203222

**Published:** 2023-10-15

**Authors:** Yuhang Miao, Tao Ding, Yang Liu, Xuezhang Zhou, Jun Du

**Affiliations:** 1College of Life Science, Ningxia University, Yinchuan 750021, China; myh6943@126.com (Y.M.); taoding113@163.com (T.D.); liuyangnihao@139.com (Y.L.); 2Key Laboratory of the Ministry of Education for the Conservation and Utilization of Special Biological Resources of Western China, Ningxia University, Yinchuan 750021, China

**Keywords:** *Candida krusei*, yeast phase, hypha phase, bovine mammary epithelial cells, apoptosis

## Abstract

**Simple Summary:**

Previous epidemiological investigations suggested that *Candida krusei* (*C. krusei*) is one of the main pathogens of mycotic mastitis in dairy cows in Yinchuan, Ningxia, China. In this study, we examined the apoptotic signaling pathways in bovine mammary epithelial cells (BMECs) induced by the *C. krusei* yeast and hypha phases using a pathogen/host cell co-culture model. The results showed that the *C. krusei* yeast phase induced the apoptosis of BMECs through a mitochondrial pathway, while the apoptosis of BMECs induced by the *C. krusei* hypha phase was reached through a death ligand/receptor pathway. In addition, both the TLR2/ERK and JNK/ERK signaling pathways were involved in the regulation of *C. krusei*-induced BMEC apoptosis. These results provide a scientific basis for developing a comprehensive prevention and treatment approach for *C. krusei* mastitis in dairy cows.

**Abstract:**

Infection with *Candida* spp. is a significant cause of bovine mastitis globally. We previously found that *C. krusei* was the main pathogen causing mycotic mastitis in dairy cows in Yinchuan, Ningxia, China. However, whether the infection of this pathogen could induce apoptosis in BMECs remained unclear. In this report, we explored the apoptosis and underlying mechanism of BMECs induced by *C. krusei* yeast and hypha phases using a pathogen/host cell co-culture model. Our results revealed that both the yeast and hypha phases of *C. krusei* could induce BMEC apoptosis; however, the yeast phase induced more cell apoptosis than the hypha phase, as assessed via electronic microscopy and flow cytometry assays. This finding was further corroborated via the measurement of the mitochondrial membrane potential (MMP) and the TUNEL test. Infection by both the yeast and hypha phases of *C. krusei* greatly induced the expression of proteins associated with cell death pathways and important components of toll-like receptor (TLR) signaling, including TLR2 and TLR4 receptors, as determined via a Western blotting assay. BMECs mainly underwent apoptosis after infection by the *C. krusei* yeast phase through a mitochondrial pathway. Meanwhile, BMEC apoptosis induced by the *C. krusei* hypha phase was regulated by a death ligand/receptor pathway. In addition, *C. krusei*-induced BMEC apoptosis was regulated by both the TLR2/ERK and JNK/ERK signaling pathways. These data suggest that the yeast phase and hypha phase of *C. krusei* induce BMEC apoptosis through distinct cell signaling pathways. This study represents a unique perspective on the molecular processes underlying BMEC apoptosis in response to *C. krusei* infection.

## 1. Introduction

*C. krusei* is a facultative saprophyte that has a wide distribution in the natural environment. It is commonly recognized as a temporary symbiotic bacterium found on mucous membranes in humans and animals [1,2,3,4]. *Candida* mastitis in dairy cows is mainly caused by infection from *Candida albicans* (*C. albicans*), but the incidence of cow mastitis caused by infection from non-*Candida albicans* such as *C. krusei* in cow mastitis has increased in recent years. An early epidemiological investigation into dairy cow mastitis over more than three years revealed that *C. krusei* was the main fungus in dairy cow mastitis, followed by *Candida parapsilosis*, in Yinchuan, Ningxia, China, while *C. albicans* was not isolated in this area [5]. Mastitis caused by *Candida* is mainly diagnosed after acute bacterial mastitis, and the pathogenic mechanism of *Candida* mastitis and the immune response of the host are currently not clear. In addition, the only option for cows with mastitis that fail to be treated with antibiotics is retirement. Therefore, *Candida* mastitis in cows is neglected in dairy production and causes huge economic losses to the local dairy farming industry in Ningxia.

*Candida* mainly causes fungal mastitis in veterinary clinical practice. The reported detection rate of *Candida* infection in the literature is higher in veterinary clinical practice than that in human clinical practice (12.2–44.5%) [6]. In this regard, Eldesouky reported the lowest detection rate of fungal mastitis in Egyptian cows, which was 12.2%. DeMarin et al. reported the highest detection rate of fungal mastitis in Brazilian cows to be 44.5%, while the detection rate of *C. albicans* was only 8.9%. Other studies reported detection rates of 12.7% [7] and 17.7% [6] in Türkiye, 17.3% [8] and 12.8% [9] in Brazil, 15.5% [10] and 21% [11] in India, and 7.5% [11] in Slovenia. In China, the detection rate was reported as 35.6% [12] in Heilongjiang Province, while it was 23.44% [5] in Yinchuan, Ningxia. The reported isolation rate of *C. krusei* in fungal mastitis was also inconsistent, with 12.2% and 15.5% in Poland [13], 15.57% in Algeria [14], 17.4% in Türkiye [6], and 34.6% [9] and 44.5% in Brazil [8]. The isolation rate reported in Heilongjiang Province was 35.6% [12], and it was 23.33%in Yinchuan, Ningxia, China [5].

Similar to *C. albicans*, *C. krusei* can produce a series of virulence factors with two-phase transformations, a key virulent characteristic of this pathogen. The yeast phase is commonly found in soils, fruits, vegetables, and fermented milk in warmer regions [15]. Unlike the yeast phase, the hypha phase is invasive and arms the bacterial strain with an ability to invade host tissues, including the mammary glands. For instance, the hypha of *C. albicans* is able to penetrate deeper tissues more efficiently than the yeast of *C. albicans* [16,17].

Bovine mammary epithelial cells (BMECs) are a population of differentiated cells found within the breast tissue of cows and are important for the architecture and function of mammary glands during milk production and secretion. In addition, BMECs also play a significant role in the immune response to bacterial infections. In this regard, they comprise the first defense barrier of the mammary tissue against the invasion of pathogenic microorganisms through recruiting circulating immune effector cells and secreting antimicrobial peptides [18,19]. 

Apoptosis is a physiological and conserved function in cells. It is essential for tissue differentiation, injury repair, and the elimination of pathogens and senescent cells. At the molecular level, it is reflected by the activation of caspase, protease, and nuclease, along with the cleavage of 180–200 bp nucleic acid and the alternation of MMP [20,21]. Previous investigations have shown that infection by *Staphylococcus aureus* and Panton–Valentine leukocidin (PVL), *Escherichia coli*, inflammation, and lipopolysaccharide (LPS) infection could induce BMEC apoptosis. For example, PVL can induce apoptosis in BMECs by elevating the expression of cleavedcaspase-3, cleavedcaspase-8, and cleavedcaspase-9 proteins [22]. BMECs can be induced to undergo inflammation and apoptosis by LPS via the NF-κB and Nrf2 signaling pathways [23,24]. In addition, *C. albicans* can induce the apoptosis of human oral mucosal keratinocytes, and the hypha phase induces more cell apoptosis than the yeast phase, resulting in a higher degree of cell apoptosis in human oral mucosal keratinocytes [2]. Mechanistically, *C. albicans* could induce the apoptosis of mouse thymocytes through the death ligand/receptor pathway and macrophage autophagy via the Akt/mTOR signaling pathway [25]. To date, *C. krusei*-induced apoptosis and its underlying mechanism in BMECs have not been reported. In this study, the apoptosis of BMECs was investigated via the infection of the cells with the yeast and hypha phases of *C. krusei* in a pathogen/host cell co-culture model. Our findings demonstrated that both the yeast and hypha phases of *C. krusei* have the capability to induce apoptosis in BMECs through different molecular signaling pathways.

## 2. Materials and Methods

### 2.1. Culture of C. krusei Yeast Phase and Hypha Phase

The reference strain ATCC 6258 of *C. krusei* was subjected to two subcultures on Sabouraud agar (Hope Bio-Technology, Qingdao, China) at a temperature of 37 °C and a CO_2_ concentration of 5% for a duration of 18–24 h. This process was carried out in order to revive the strain and ensure the purity of cultures. Subsequently, a solitary colony was transferred to a YPD liquid medium (Hope Bio-Technology) and *C. krusei* yeast phase cells were cultivated at a temperature of 37 °C in a shaking incubator operating at a speed of 120 rpm for a duration of 12–18 h. When the cells reached 5 × 10^7^/mL, the medium was removed carefully by centrifuging the yeast phase cells at 5000 rpm for 5 min. *C. krusei* hypha phase cells were harvested by culturing in RPMI-1640 medium supplemented with 10% FBS (Gibco, Thermo Fisher Scientific, Grand Island, NY, USA) for 4 hat 37 °C in a 150 rpm rotating incubator.

### 2.2. Cell Culture

The MAC-T cell line, derived from BMECs, was acquired from Shandong Agricultural University in Tai’an, China [26,27]. BMECs were cultured in a 90 mm petri dish with DMEM (Hyclone, Logan, UT, USA) supplemented with 10% fetal bovine serum (FBS) and 1% penicillin/streptomycin. The cells were subjected to incubation under controlled conditions of temperature (37 °C) and carbon dioxide concentration (5%) in a humidified atmosphere.

### 2.3. Infections of BMECs with C. krusei

BMECs were grown to the logarithmic growth phase and then seeded overnight in 6-well plates (5 × 10^5^ cells/well). For use in the assays, the density of the *C. krusei* yeast phase and hypha phase was adjusted to 5 × 10^6^ per mL by turbidimetry at a wavelength of 600 nm. A volume of 100 µL of each *C. krusei* yeast and hypha phases suspensions was inoculated in technical triplicate into 6-well plates. BMECs were infected with either the yeast or hypha phase of the *C. krusei* ATCC6258 strain for 4 or 8 h at a multiplicity of infection of 1 (MOI = 1). Cells were rinsed three times in sterile PBS before being harvested for analysis with 3 replicates per group. Uninfected BMECs were used as the negative control. BMECs infected with *C. krusei* yeast phase for 4 and 8 h were designated as the J4 and J8 group, respectively. BMECs infected with *C. krusei* hypha phase cells for 4 and 8 h were termed as the S4 and S8 group, respectively. All experiments were performed in technical triplicate and biologically repeated with BEMCs from 3 donor cows.

### 2.4. Transmission Electron Microscopy

The cells were subjected to fixation in 3% glutaraldehyde solution for a duration of 24 h. Following fixation, the cells were rinsed with PBS and subsequently post-fixed in 1.0% osmium tetroxide solution. Subsequently, the samples underwent dehydration using a sequential immersion in alcohol solutions of varying concentrations (30%, 50%, 70%, 80%, 90%, 95%, and 100%). Following critical-point drying, the samples were immersed in a mixture of glycide-ether and propylene oxide. After the polymerization of propylene oxide, ultra-thin sections were obtained using an ultramicrotome (Leica EM, Oskar Barnack, Germany). These sections were then stained with a 1% solution of uranyl acetate followed by lead citrate. The morphology of samples was observed and imaged using a JEM-1400 FLASH transmission electron microscope (JEOL, Tokyo, Japan).

### 2.5. Analysis of Apoptosis/Necrosis with AV/PI Double Staining

A Dead Cell Apoptosis Kit with Alexa Fluor^®^488 annexin V and PI for Flow Cytometry (Invitrogen, Thermo Fisher Scientific, Waltham, MA, USA) was used to detect apoptosis in BMECs. Briefly, cells were seeded for 24 h at 37 °C and 5% CO_2_ in a 60 mm culture dish. After infection, the cells were separated, centrifuged for 5 min at 1000 rpm, and rinsed twice with ice-cold PBS. The cells were then resuspended in the binding buffer at a concentration of 1 × 10^6^ cells/mL, and 100 µL of the cell suspension was stained with 5 µL of Annexin V-FITC and 5 µL of PI for 15 min at room temperature in the dark. Apoptosis was analyzed using a Sysmex CyFlow Cube8 (PartecGmbh, Goerlitz, Sachsen, Germany).

### 2.6. MMP Assay

MitoProbe™ JC-1 assay reagent (Invitrogen, Thermo Fisher Scientific, Waltham, MA, USA) was employed for the detection of MMP loss in BMECs. Following infection, cells were collected, rinsed twice with ice-cold PBS, and stained with JC-1 per the manufacturer’s instructions. A Sysmex CyFlow Cube8 (PartecGmbh, Goerlitz, Sachsen, Germany) was used to analyze the stained cells.

### 2.7. Measurement of Intracellular ROS

The intracellular reactive oxygen species (ROS) of BMECs were assessed using the DCFH-DA fluorescent probe (KeyGENBioTECH, Nanjing, China). The cells were harvested and subsequently subjected to two rounds of washing with ice-cold PBS, before they were stained with DCFH-DA at a concentration of 10 µM in DMEM for a duration of 30 min at a temperature of 37 °C in dark. The cells were rinsed twice with PBS after the medium was removed. The fluorescence intensity of the DCF was then quantified using a Sysmex CyFlow Cube8 (PartecGmbh, Goerlitz, Sachsen, Germany).

### 2.8. TdT-Mediated dUTP Nick End Labeling (TUNEL) Assay

Following the manufacturer’s instructions, a TUNEL assay was performed to detect apoptotic BMECs with DNA fragmentation using the TUNEL Bright Red Apoptosis Detection Kit (Vazyme Biotech, Nanjing, China).

### 2.9. Western Blot Analysis

The BMECs were cultured in 6-well culture plates (Corning Incorporated, Corning, NY, USA) at an inoculation density of 5 × 10^5^ cells per well in medium without the addition of FBS. They were then exposed to *C. krusei* yeast phase and hypha phase cells at MOI = 1 for the specified durations. Subsequently, the lysates of BMECs were obtained through the utilization of a total protein extraction kit manufactured by KeyGen Biotech, Nanjing, China. The protein concentrations were determined utilizing the BCA Protein Assay Kit manufactured by KeyGen Biotech, Nanjing, China. A protein sample of fifteen micrograms was subjected to sodium dodecyl sulfate–polyacrylamide gel electrophoresis (SDS-PAGE) using a 12% gel. Subsequently, the separated proteins were deposited onto polyvinylidene fluoride (PVDF) membranes obtained from Epizyme Biotech in Shanghai, China. Following a 2 h period of blocking in a solution of TBS containing 0.05% Tween 20 and 5% non-fat milk, the membranes were subsequently subjected to overnight incubation at 4 °C with antibodies specific to the proteins of interest: primary rabbit antibodies against β-actin (1:1000), anti-total ERK1/2 (1:1000), anti-total JNK (1:1000), anti- phospho-ERK1/2 (1:1000), anti-phosphor-JNK (1:1000), anti-cleaved-caspase-3 (1:1000), anti-cleaved-caspase-8 (1:1000), anti-cleaved-caspase-9 (1:1000), anti-caspase-3 (1:1000), anti-caspase-8 (1:1000), anti-Bim (1:1000), anti-TLR2 (1:1000), anti-TLR4 (1:1000), anti- MYD88 (1:1000), anti-FOXO4 (1:1000), anti-phospho-FOXO4 (1:1000), anti-APAF1 (1:1000), anti-Fas ligand (1:1000) (all from Bioss, Beijing, China); anti-caspase-9 (1:1000) and anti-Bax (1:1000) (from Proteintech, Wuhan, China); anti-total P38 (1:1000) (from Abcam, UK);anti-phospho-P38 (1:1000) (from ThermoFisher, Waltham, MA, USA); and anti-Bcl-2 (1:1000) (Santa Cruz Biotech, Santa Cruz, CA, USA). In order to perform suitable secondary antibody incubations, we utilized horse radish peroxidase (HRP)-conjugated goat anti-mouse or goat anti-rabbit antibodies (Proteintech, Wuhan, China) at a dilution of 1:5000. An enhanced chemiluminescence (ECL) Western Blot Detection System (Amersham Imager 600RGB, GE Amersham, Fairfield, CT, USA) was utilized to visualize the binding of antibodies and target proteins. Actin was used as house-keeping protein for loading control.

### 2.10. Treatment with Inhibitors

For the inhibition of caspase-3, caspase-8, caspase-9, ERK, and JNK, TLR2, BMECs were pre-incubated with 30 µM Caspase-3/7 inhibitor I (A1925; ApexBio Technology, Houston, TX, USA) for 24 h; 150 µM Z-IETD-FMK (B3232; ApexBio Technology, Houston, TX, USA) for 24 h; 40 µM Z-LEHD-FMK (B3233; ApexBio Technology, Houston, TX, USA) for 24 h; 20 µM ERK inhibitor (C4096; ApexBio Technology, Houston, TX, USA) for 2 h; 100 µM JNK-IN-7 (A3519; ApexBio Technology, CA, USA) for 2 h; or 100 µM C29 (HY-100461; MedChemExpress, Monmouth Junction, Trenton, NJ, USA) for 2 h prior to infection. DMSO was served as the vehicle control for the inhibitor, and the untreated BMECs served as naive control. BMECs pretreated with inhibitors were designated as the YC group, respectively.

### 2.11. Statistical Analysis

The data were presented as the standard error of the mean ± standard deviation (SD), and the means were compared using one-way analysis of variance (ANOVA) with GraphPad Prism (GraphPad 8.0.1 Software Inc., San Diego, CA, USA). The graphs were made using GraphPad Prism 8.0.1 software. The experiments were conducted in triplicate, with each trial being replicated three times for technical accuracy. A significance level of *p* < 0.05 was used to determine statistical significance.

## 3. Results

### 3.1. Infections of C. krusei Yeast Phase and Hypha Phase Induced Morphological Changes in Apoptosis in BMECs

The apoptosis was first evaluated by assessing morphological changes in BMECs infected with *C. krusei* yeast phase and hypha phase using a transmission electron microscope (TEM). Normal cell morphology and structure with irregular polygon nuclei and a uniform distribution of euchromatin were observed in the uninfected control cells. The cytoplasm of the uninfected cells exhibited distinct cellular features, such as a continuous nucleolus, intact entire nuclear membrane, mitochondria, Golgi apparatus, rough endoplasmic reticulum, ribosomes, and other organelles (Figure 1A,B). In contrast, BMECs that were infected by the *C. krusei* yeast phase (Figure 1C–F) or hypha phase (Figure 1G–J) showed apoptotic morphology with nuclear shrinkage, chromatin aggregation, swollen mitochondria, autophagosomes, vacuoles, and a small amount of lipid droplets in the cytoplasm at both 4 and 8 h after the infection (Figure 1C–J).

### 3.2. BMECs Infected with C. krusei Exhibit an Apoptotic Phenotype

To further determine the apoptotic phenotype of BMECs in response to *C. krusei* infection, cells were first stained with fluorescent dye AV/PI, and phenotypes of apoptosis and necrosis were tested with flow cytometry (Figure 2A,B). Compared with the control group, the apoptosis rate of the *C. krusei* yeast phase-infected cells was significantly increased at 2 h, 4 h, 6 h, 8 h, and 10 h (*p* < 0.001); the apoptotic rate was the highest at 8 h. The apoptosis rate of the *C. krusei* hypha phase-infected cells was significantly increased at 2 h and 4 h (*p* < 0.05) and at 6 h and 8 h (*p* < 0.001). There was no significant difference in the rate of apoptosis at 10 h (Figure 2A and Figure 3A). The necrosis rate of the *C. krusei* yeast phase-infected cells was significantly increased at 2 h compared to the control group (*p* < 0.05); the necrosis rate was significantly increased at 8 h and 10 h (*p* < 0.001). There was no significant difference between the 4 h and 6 h necrosis rates, except for 4 h. The necrosis rate of *C. krusei* hypha phase-infected cells was significantly increased in all treated groups (*p* < 0.05 and *p* < 0.001), and the highest necrosis rate was at 6 h (Figure 2B and Figure 3B). Of note, the yeast phase of *C. krusei* induced more apoptosis of BMECs compared to the hypha phase of bacteria at all examined time points (Figure 3A). In addition, infection with the *C. krusei* yeast phase and hypha phase significantly increased the necrotic frequency of BMECs (Figure 3B). Interestingly, the induction of cell necrosis was dynamically changed; less cell necrosis was observed at 4 and 6 h post *C. krusei* yeast phase infection (Figure 3B). Unlike infection with the yeast phase of *C. krusei*, which induced more cell apoptosis than that with the hypha phase, infection with the hypha phase of *C. krusei* caused more necrotic cells than the yeast phase (Figure 3A,B). The *C. krusei*-induced BMEC apoptosis was further confirmed by MMP and ROS in BMECs infected by the *C. krusei* yeast phase and hypha phase. According to the scatter and statistical plots of MMP (Figure 2C and Figure 3C), compared with the control group, the MMP of BMECs infected by the *C. krusei* yeast phase and hypha phase decreased significantly. According to the data collated in the schematic diagram and statistical diagram (Figure 2D and Figure 3D), compared with the control, the accumulation of ROS did not increase significantly. Furthermore, the TdT-mediated dUTP Nick End Labeling(TUNEL) assay revealed more abundant FITC-labeled apoptotic cells in BMECs infected with *C. krusei* at 4 h and 8 h after the infection, in comparison to the control group (Appendix A). Consistent with the morphological and cytometry findings, molecular analysis using a Western blotting assay also showed significantly increased expression of pro-apoptosis proteins, including Bim, APAF1, death ligand FasL, cleavedcaspase-9, cleavedcaspase-8, cleavedcaspase-3, p-FOXO4, and Bax, coupled with a significantly decreased level of the anti-apoptosis protein Bcl-2, in BMECs infected with the yeast phase or hypha phase of *C. krusei* (Figure 4).

### 3.3. Distinct Apoptosis Signaling Pathways in BMECs Infected with the Yeast Phase and Hypha Phase of C. krusei

To examine the intrinsic signaling pathways that mediated BMEC apoptosis in response to infection with the *C. krusei* yeast and hypha phases, the effect of caspase-3/7 inhibitor I, Z-LEHD-FMK, and Z-IETD-FMK on the *C. krusei*-induced BMEC apoptosis was analyzed by assessing the change incaspase-3, caspase-8, and caspase-9 proteins. There sults showed that the pretreatment ofcaspase-3/7 inhibitor I significantly decreased the expression of cleavedcaspase-3 in BMECs infected with *C. krusei* (Figure 5A). At the same time, the Z-IETD-FMK pretreatment significantly downregulated cleavedcaspase-3 expression and cleavedcaspase-8 expression in BMECs infected with the hypha phase of *C. krusei*, but it had no effect on the cleavedcaspase-8 expression in BMECs infected with the yeast phase of *C. krusei* (Figure 5B). In contrast, Z-LEHD-FMK preconditioned the BMECs, so infection with the yeast phase of *C. krusei* significantly decreased cleavedcaspase-3 expression and cleavedcaspase-9 expression. However, the BMECs infected with the hypha phase of *C. krusei* did not undergo any significant changes in the expression of cleavedcaspase-3 or cleavedcaspase-9 (Figure 5C). 

### 3.4. TLR/ERK Signaling Is Activated in BMECs in Response to C. krusei Infection

The results of measuring the expression level and activity of toll-like receptor-related proteins in cells showed that the expression of MYD88, IRF3, IRF5, and TLR2 could be significantly enhanced by both the *C. krusei* yeast phase and hypha phase, while only the hypha phase significantly enhanced the expression of TLR4 (Figure 6). To verify the potential role of TLR2 in the apoptosis of BMECs induced by *C. krusei* yeast and hypha phases, we analyzed the effect of C29 pretreatment on cell damage. The results demonstrated that C29 pretreatment substantially inhibited the infection-induced expression of TLR2 and reduced the expression of cleavedcaspase-3 and ERK, but could alter the expression of p-P-38 (Figure 7 and Figure 8). 

### 3.5. JNK and ERK Pathways Are Involved in BMEC Apoptosis in Response to C. krusei Infection

Western blotting analysis was employed to quantify the activity and expression levels of proteins associated with the JNK and ERK signaling pathways (Figure 9). However, there was no observed impact on the expression of P-38 and p-P-38. To ascertain the precise involvement of the ERK and JNK signaling pathways in BMEC apoptosis induced by the *C. krusei* yeast phase and hypha phase, we analyzed the effects of ERK inhibitor and JNK-in-7 pretreatment on cells, respectively. The results showed that ERK inhibitor pretreatment significantly inhibited p-ERK and cleavedcaspase-3 expression (Figure 10A). The pretreatment with JNK-in-7 significantly inhibited p-JNK and cleavedcaspase-3 expression (Figure 10B).

## 4. Discussion

*C. krusei* is a pathogen whose infection can cause several serious diseases in both humans and animals [28]. *Candida* mastitis primarily arises from an infection caused by the pathogen of *C. albicans* in dairy cows. The prevalence of bovine mastitis in dairy cows attributed to non-albicans *Candida*, such as *C. krusei*, has shown an increasing trend in recent years [14,29,30]. *C. krusei* can colonize on the surface of instruments such as milkers and perfusion catheters and enter the udder of cows through contaminated perfusion catheters and syringes, causing mammary inflammation. Owing to its inherent resistance to fluconazole, together with biofilm formation, secretase ability, and strong cell surface hydrophobicity, mammary tissues are more susceptible to *C. krusei* when the balance of the microbial ecosystem in animals is disrupted by antibiotic treatment [1].

A number of studies have demonstrated that infection with *C. albicans* can induce apoptosis in human monocytes, macrophages, human oral mucosal epithelial keratinocytes, and human dendritic cells [31,32,33,34]. To date, only the cell wall mannan of *C. krusi* has been reported to mediate dendritic cell apoptosis [35]. In this study, AV/PI double staining results showed that both the yeast and hypha phases of *C. krusei* could induce the apoptosis and necrosis of BMECs, but apoptosis was the predominant cell death type caused by the two phases. Interestingly, infection with the *C. krusei* hypha phase caused more necrotic cell death than that with the yeast phase of the bacteria. These results suggest that the hypha phase is more likely to cause necrosis while inducing the apoptosis of BMECs.

Another interesting finding from the TUNEL assay is that BMECs infected with either the yeast or hypha phase of *C. krusei* for 4 h and 8 h exhibited DNA fragmentation, while the yeast phase induced significantly higher DNA fragmentation than the hypha phase. Ultrastructural observation showed that apoptosis and autophagy occurred in cells 4 h and 8 h after infection with the *C. krusei* yeast phase and hypha phase. These results indicate that both the yeast and hypha phases of *C. krusei* were able induce the apoptosis of BMECs, but the degree was different. The apoptosis induced by the yeast phase was higher than that of the hypha phase. The apoptosis rate of the hypha phase was substantially higher than that of the yeast phase in *C. albicans*-infected macrophages, which was inconsistent with our experimental results, which may be related to the secretion of more aspartic protease in the hypha phase of *C. albicans* [36,37]. The filamentation of *Candida* plays a pivotal role in the adhesion, invasion, and damage of epithelial and endothelial cells [38,39,40]. The top of the hypha is the site of secretion of aspartyl proteinase, which can destroy the cell membrane of the host cell and promote a *Candida* invasion of tissue [41,42]. In addition, the growth of the hypha also increases the resistance of *C. albicans* to phagocytes. In vitro experiments have shown that the yeast phase of *C. albicans* could be transformed into a hypha phase after being phagocytosed by macrophages, and could puncture macrophages and cause their death [43]. However, the yeast-to-hypha transition shows an ability to escape from the phagosome at the later stage of the infection process [44,45,46]. It was found that both *C. krusei* yeast phase and hypha phase infection could lead to a chronic inflammatory reaction in a rat mammary gland in a rat mastitis model generated by the infection of *C. krusei* yeast phase and hypha phase. In addition, cytokines TNF-α, IL-1β, IL-6, IL-8, and IL-18, along with serum albumin and transferrin, were also found to be involved in the inflammatory damage caused by *C. krusei* yeast phase and hypha phase infection in rat mammary breast, and the inflammatory damage caused by the hypha phase was more severe than that caused by the yeast phase (unpublished data). It is also speculated that the *C. krusei* hypha phase may secrete hydrolase and cause more necrosis of BMECs, ultimately promoting the development of mastitis.

In addition, the examination of cellular ultrastructure, MMP, and molecular analysis utilizing various inhibitors of apoptosis pathways indicated that the apoptosis of BMECs induced by the *C. krusei* yeast phase was primarily involved in the mitochondrial pathway. Conversely, the apoptosis of BMECs induced by the *C. krusei* hypha phase was found to be associated with the death ligand/receptor pathway. Some studies have reported that *C. albicans* could induce the apoptosis of mouse thymocytes through the death ligand/receptor pathway [47]. In contrast, *C. krusei* induced the apoptosis of BMECs through a mitochondrial pathway and death ligand/receptor pathways. Moreover, the expression of Bim and p-FOXO4 was also significantly increased in BMECs infected with *C. krusei*, suggesting that *C. krusei* could induce BMEC apoptosis through the p-FOXO4/Bim signaling pathway, and the precise molecular mechanism requires investigation by subsequent experimental investigations [48].

The innate immune response of BMECs is initiated by pattern recognition receptors (PRRs) in response to pathogen-associated molecular patterns (PAMPs) [49]. Toll-like receptors (TLRs) represent the most extensively studied type of PRR. The aforementioned receptors hold significant importance within the innate immune system of the mammary gland, namely, in the realm of pattern recognition. The innate and acquired immune response is significantly influenced by its pivotal role [50,51,52]. The first PRRs discovered to recognize *C. albicans* were found to belong to the TLR family [53]. A recent study showed that TLR2 may contribute to protection against the colonization and endogenous invasion by *C. albicans* [54]. Accumulated evidence has indicated that *Candida glabrata* infection can induce apoptosis in the airway epithelial cells of mice, and the TLR2 receptor could mediate the NF-κB signaling pathway in order to stimulate the secretion of downstream inflammatory factors, which play a key role in fungal recognition and antifungal immunity [55]. These studies indicated that PRRs are different when *Candida* infects a variety of cells. Indeed, the expression of MyD88, IRF3, IRF5, and TLR2 was significantly increased in BMECs infected by both the yeast phase and the hypha phase of *C. krusei*, while the expression of TLR4 was significantly increased by the *C. krusei* hypha phase. These findings are consistent with previous findings that both the yeast phase and the hypha phase of *C. albicans* can induce TLR2 and TLR4 upregulation in T cells and human intestinal epithelial cells [56]. It was speculated that both the yeast phase and hypha phase of *C. krusei* could activate TLR2 in BMECs through the MYD88-dependent signaling pathway, while the activation of TLR4 in BMECs through the MyD88-independent signaling pathway might also be triggered by the hypha phase. This was in concordance with results of *C. krusei* infection in dendritic cells [35]. The results of Western blotting analysis related to MAPK signaling pathway proteins showed that both the yeast phase and hypha phase of *C. krusei* could increase the expression of ERK, p-ERK, JNK, and p-JNK, but not P38 and p-P38. However, previous studies demonstrated that the *C. albicans* hyphal burden could activate the MAPK signaling pathway during the infection. In this regard, MAPK could activate P38, leading to c-Fos activation [57,58]. This is inconsistent with our results and suggests that different host cells might undergo infection with different *Candida* species through different signaling pathways. From our results, we speculated that the yeast and hypha phases of *C. krusei* could activate the ERK and JNK pathways but not the P38 kinase pathways in BMECs. Together with the finding from the pretreatment experiments with inhibitors of TLR2, ERK, and JNK pathways, the results presented in this study indicate that the signaling pathways involved in TLR2/ERK and JNK signaling play significant roles in the regulation of apoptosis in BMECs.

## 5. Conclusions

The *C. krusei* yeast phase and hypha phase can both induce apoptosis in BMECs. The apoptosis of BMECs induced by the *C. krusei* yeast phase was greater than that of the *C. krusei* hypha phase, while the hypha phase of *C. krusei* caused more necrosis than the yeast phase. The *C. krusei* yeast phase and hypha phase could activate TLR2 in BMECs through MYD88-dependent signaling, while the hypha phase of *C. krusei* could activate TLR4 through the MyD88-independent signaling pathway in BMECs. Mechanistically, the apoptosis of BMECs induced by the *C. krusei* yeast phase was mainly through the mitochondrial pathway of apoptosis, while apoptosis induced by the *C. krusei* hypha phase was mainly regulated by the death ligand/receptor pathway. Furthermore, the apoptosis of BMECs in response to *C. krusei* infection was regulated by both the TLR2/ERK and JNK/ERK signaling pathways.

## Figures and Tables

**Figure 1 animals-13-03222-f001:**
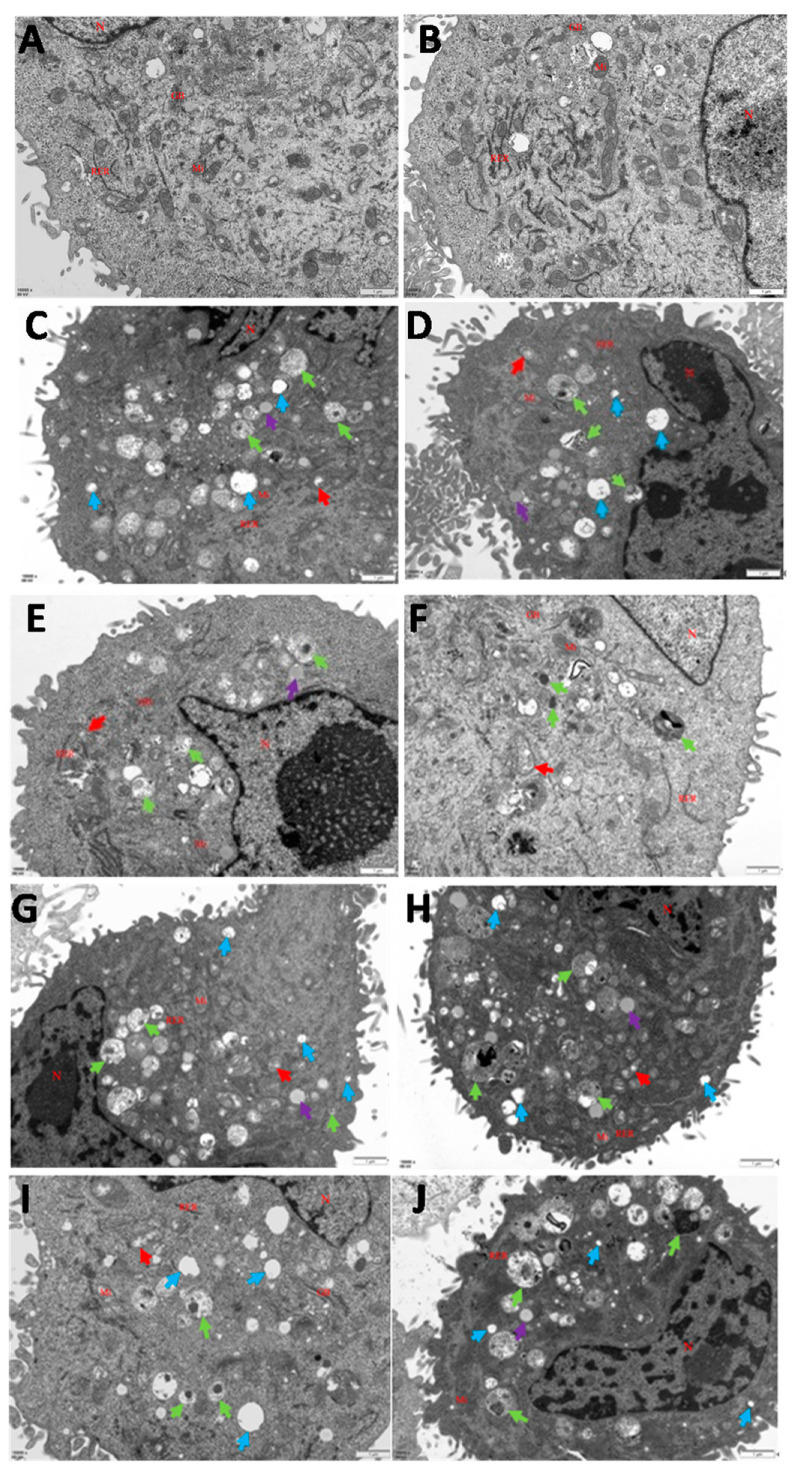
Transmission electron photomicrographs showing ultrastructural pathological alterations in BMECs. (**A**,**B**) The photomicrographs of control group. (**C**,**D**) Photomicrographs of 4 h *C. krusei* yeast phase-infected cells. (**E**,**F**) Photomicrographs of 8 h *C. krusei* yeast phase-infected cells. (**G**,**H**) Photomicrographs of 4 h *C. krusei* hypha phase-infected cells. (**I**,**J**) Photomicrographs of 8 h *C. krusei* hypha phase-infected cells. Note: magnification ×15,000; red shears indicate mitochondrial swelling; green shears indicate autophagy; blue shears indicate vacuoles; purple shears indicate drops of grease.

**Figure 2 animals-13-03222-f002:**
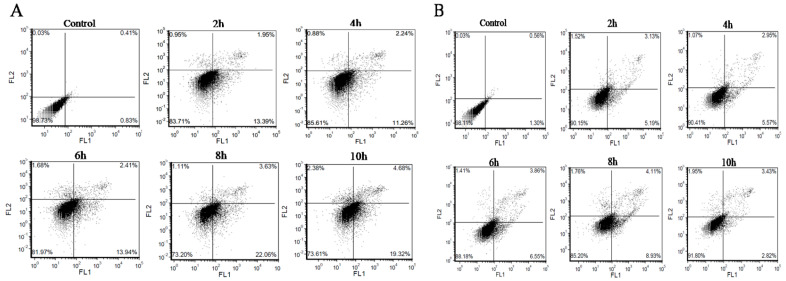
BMECs apoptosis analyzed by flow cytometry. (**A**) The induction of apoptosis and necrosis in BMECs by the yeast phase of *C. krusei* was evaluated using flow cytometry, employing annexin V/propidium iodide (PI) double labelling. (**B**) The induction of apoptosis and necrosis in BMECs by the hypha phase of *C. krusei* was assessed by flow cytometry, specifically through the application of annexin V/propidium iodide (PI) double labelling. (**C**) MMP of BMECs induced by *C. krusei* yeast phase and hypha phase via flow cytometry with JC-1 fluorescence probe. Positive cells exhibiting red fluorescence are observed in the upper quadrant, while negative cells are observed in the lower quadrant. (**D**) Intracellular ROS generation of BMECs induced by *C. krusei* yeast phase and hypha phase via flow cytometry with DCFH-DA fluorescence probe.

**Figure 3 animals-13-03222-f003:**
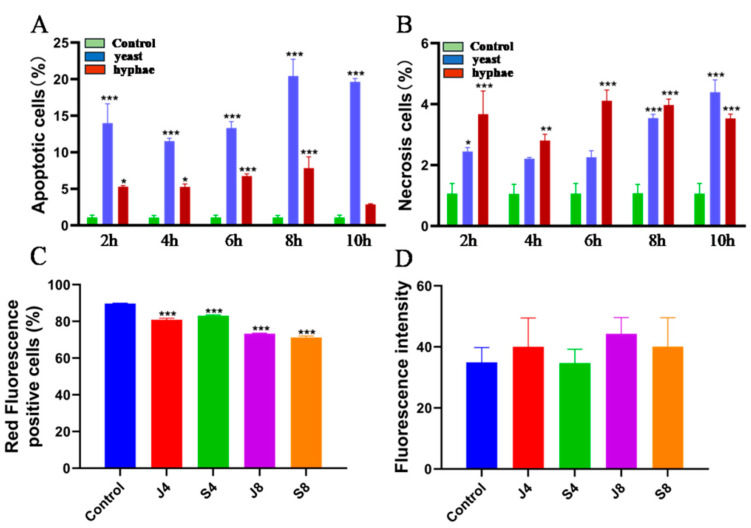
*C. krusei* yeast phase- and hypha phase-induced apoptosis of BMECs. (**A**) Apoptosis rate of BMECs. (**B**) Necrosis rate of BMECs. (**C**) The rate of MMP-positive BMECs. (**D**) The rate of ROS-positive BMECs. The data are presented in the form of mean ± SD derived from three separate and independent studies. The statistical significance levels are denoted as follows: * *p* < 0.05,** *p* < 0.01, and *** *p* < 0.001, in comparison to the control group.

**Figure 4 animals-13-03222-f004:**
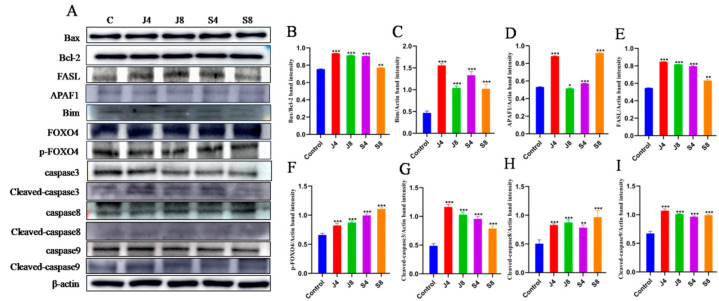
Western blot analysis of apoptosis-related proteins in BMECs incubated with *C. krusei* yeast phase and hypha phase. (**A**) Representative blots of Western blotting assays of Bax, Bcl-2, Bim, APAF1, FasL, FOXO4, p-FOXO4, caspase-8, cleavedcaspase-8, caspase-9, cleavedcaspase-9, caspase-3, and cleavedcaspase-3 proteins at 4 and 8 h after *C. krusei* yeast phase and hypha phase infection in BMECs. (**B**) Quantitative graph of relative expression of Bax/Bcl-2 proteins. (**C**) Semi-quantitative graph of Bim protein expression levels. (**D**) Quantitative map of APAF1 protein expression levels. (**E**) Semi-quantitative graph of FasL protein expression levels. (**F**) Quantitative graph of relative expression of p-FOXO4 protein. (**G**) Semi-quantitative graph of cleavedcaspase-3 protein. (**H**) Semi-quantitative graph of relative expression of cleavedcaspase-8 protein. (**I**) Semi-quantitative graph of relative expression of cleavedcaspase-9 protein. β-actin served as a control in this experiment. Results from three independent experiments are presented as mean ± SD. Compared to the control group, * *p* < 0.05, ** *p* < 0.01 and *** *p* < 0.001 were significant. Original western blot figures in Appendix A.

**Figure 5 animals-13-03222-f005:**
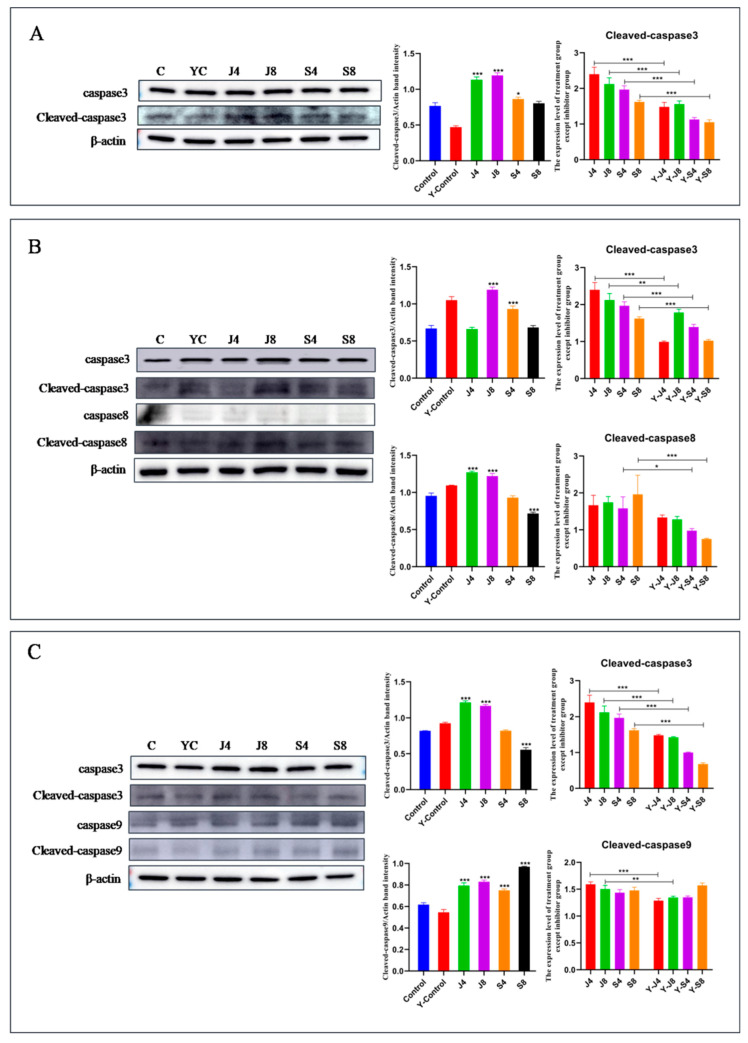
Analysis of the apoptotic pathway of BMECs induced by *C. krusei* yeast phase and hypha phase. (**A**) The expression of apoptosis proteins in BMECs pretreated with caspase-3/7 inhibitor I. (**B**) The expression of apoptosis proteins in BMECs pretreated with Z-IETD-FMK. (**C**) The expression of apoptosis proteins in BMECs pretreated with Z-LEHD-FMK. β-actin served as a control in this experiment. Results of three independent experiments are presented as mean ± SD. Compared to the control group, * *p* < 0.05, ** *p* < 0.01, and *** *p* < 0.001 were significant.

**Figure 6 animals-13-03222-f006:**
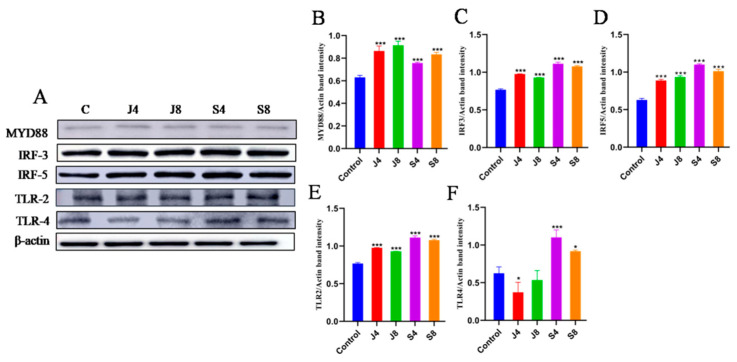
Western blot analysis of Toll-like receptor-related proteins in BMECs infected with *C. krusei* yeast phase and hypha phase. (**A**) The representative blots of Western blotting assays of MYD88, IRF3, IRF5, TLR2, and TLR4 proteins of BMECs at 4 and 8 h after *C. krusei* yeast phase and hypha phase infection. (**B**) Semi-quantitative graph of the expression of MYD88 protein. (**C**) Semi- quantitative graph of the expression of IRF3 protein. (**D**) Semi-quantitative graph of the expression ofIRF5 protein. (**E**) Semi-quantitative graph of the expression of TLR2 protein. (**F**) Semi- quantitative graph of the expression of TLR4 protein. β-actin served as a control in this experiment. Results of three independent experiment are presented as mean ± SD. Compared to the control group, * *p* < 0.05 and *** *p* < 0.001 were significant.

**Figure 7 animals-13-03222-f007:**
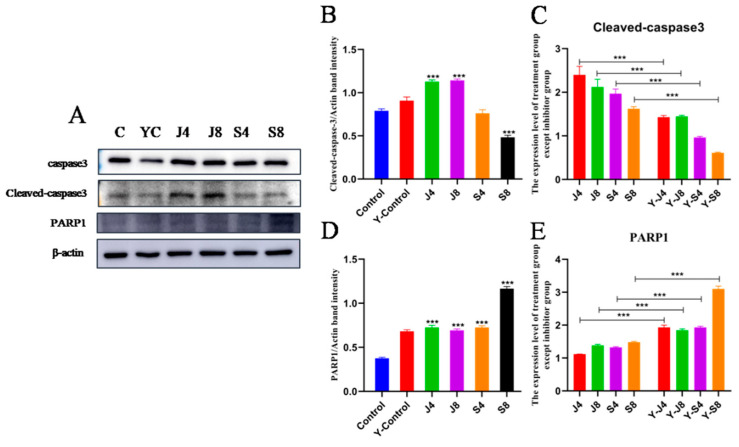
Apoptosis protein expression assay of BMECs pretreated with C29. (**A**) Representative blots of Western blotting analyses of caspase-3, cleavedcaspase-3, and PARP1 protein in BMECs pretreated with C29 at 4 and 8 h after *C. krusei* yeast phase and hypha phase infection. (**B**) Semi-quantitative graph of the expression of cleavedcaspase-3 protein. (**C**) Semi-quantitative graph of the expression of cleavedcaspase-3 protein, excluding the no-inhibitor-treatment groups. (**D**) Semi-quantitative graph of PARP1 protein expression. (**E**) Semi-quantitative graph of PARP1 expression, excluding the no-inhibitor-treatment groups. β-actin served as a control in this experiment. Results of three independent experiments are presented as mean ± SD. Compared to the control group, *** *p* < 0.001 was significant.

**Figure 8 animals-13-03222-f008:**
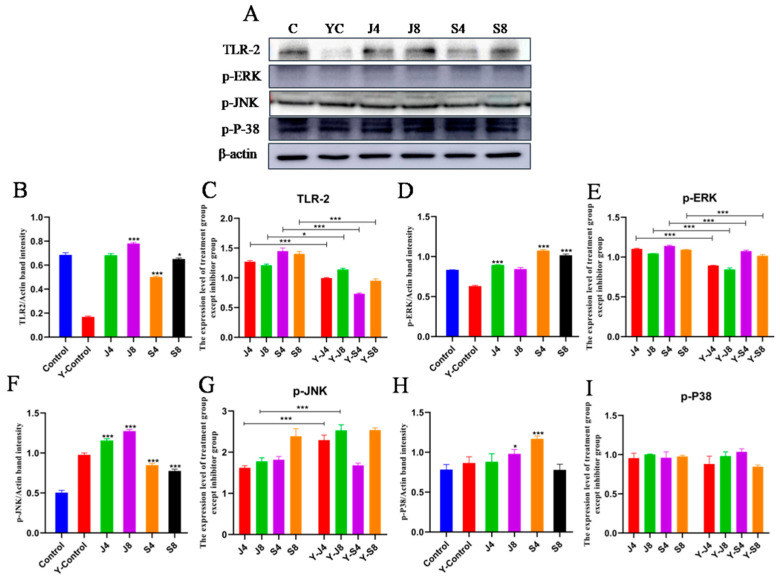
Expression of MAPK signaling pathway-related proteins in BMECs pretreated with C29. (**A**) Representative blots of Western blot assays of p-ERK, p-JNK, p-P-38, and TLR2 proteins in BMECs pretreated with C29 at 4 and 8 h after *C. krusei* yeast phase and hypha phase infection. (**B**) Semi-quantitative graph of the expression of TLR2 protein. (**C**) Semi-quantitative graph of the expression of TLR2 protein in treatment groups. (**D**) Semi-quantitative graph of p-ERK protein expression levels. (**E**) Semi-quantitative graph of the expression of p-ERK in treatment groups. (**F**) Semi-quantitative graph of the expression of p-JNK protein. (**G**) Semi-quantitative graph of the expression of p-JNK protein in treatment groups. (**H**) Semi-quantitative graph of the expression of p-P-38 protein. (**I**) Semi-quantitative graph of the expression of p-P-38 protein in treatment groups. β-actin served as a control in this experiment. Results of three independent experiments are presented as mean ± SD. Compared to the control group, * *p* < 0.05 and *** *p* < 0.001 were significant.

**Figure 9 animals-13-03222-f009:**
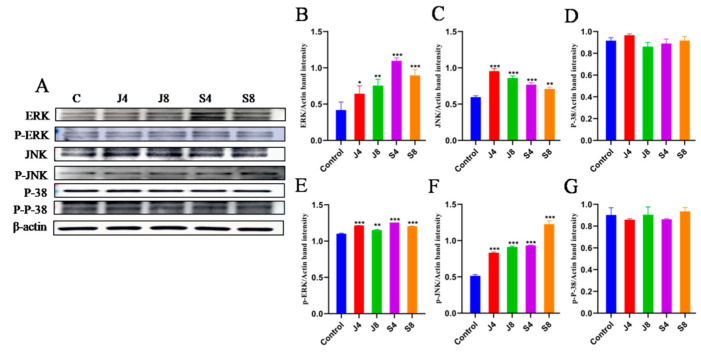
Western blotting analysis of the expression of MAPK signaling pathway-related proteins in BMECs incubated with *C. krusei* yeast phase and hypha phase. (**A**) Representative blots of Western blotting assays of the expression of ERK, JNK, P-38, p-ERK, p-JNK, and p-P-38 proteins in BMECs at 4 and 8 h after *C. krusei* yeast phase and hypha phase infection. (**B**) Semi-quantitative graph of the expression of ERK protein. (**C**) Semi-quantitative graph of the expression of JNK protein. (**D**) Semi-quantitative graph of the expression of P-38 protein. (**E**) Semi-quantitative graph of the expression of p-ERK protein. (**F**) Semi-quantitative graph of the expression of p-JNK protein. (**G**) Semi-quantitative graph of the expression ofp-P-38 protein. β-actin served as a control in this experiment. Results of three independent experiments are presented as mean ± SD. Compared to the control group, * *p* < 0.05, ** *p* < 0.01, and *** *p* < 0.001 were significant.

**Figure 10 animals-13-03222-f010:**
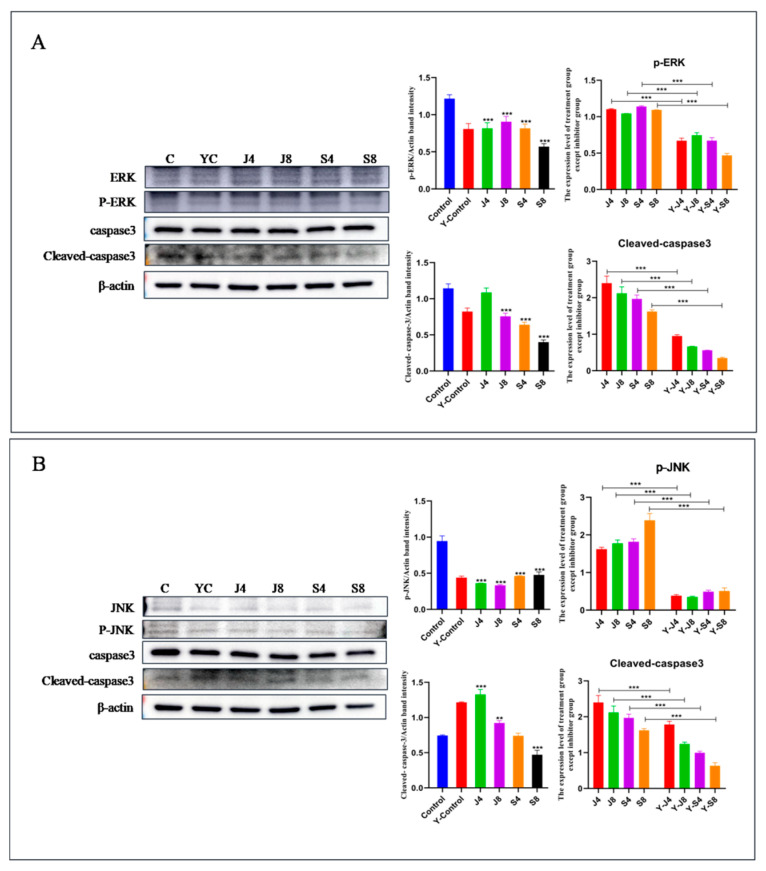
The expression of pro-apoptosis proteins in BMECs pretreated with inhibitor. (**A**) Expression of apoptosis proteins of BMECs pretreated with ERK inhibitor. (**B**) Expression of apoptosis proteins of BMECs pretreated with JNK-IN-7. Left panels show representative blots of Western blotting assay; right panels show the semi-quantitative graphs of blots shown in left of (**A**,**B**). β-actin served as a control in this experiment. Results of three independent experiments are presented as mean ± SD. Compared to the control group, ** *p* < 0.01 and *** *p* < 0.001 were significant.

## Data Availability

The datasets generated for this study are available on request to the corresponding author.

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
