# Peer review of "The Yeast and Hypha Phases of Candida krusei Induce the Apoptosis of Bovine Mammary Epithelial Cells via Distinct Signaling Pathways"

_animals, 2023, doi:10.3390/ani13203222_

Round 1
Reviewer 1 Report
The present article tries to uncover the the signalling pathways responsible of BMECs necrosis/apoptosis in the presence of different phases of Candida krusei. Indeed it's an interesting topic to discuss and the possible implications on animal husbandry and breeding emerging from this type of studies could be really relevant.
even though the topic is extremely relevant and interesting the article itself has some minor flaws that need to be addressed in order to publish it.
Line 69: Use the abbreviation PVL only after writing the word in full. Please make sure to check this issue throughout all the text.
Line 97: Use only BMECs, you already have written the word in full previously in the text. Please make sure to check this kind of issue throughout all the text.
Line 233-235: Please rephrase the sentence, it's not clear what you are trying to say.
Please specify what stands for YC in figure 4, 6, 7 and 9 since this additional control in never mentioned in the main text.
The authors should improve the discussion section with literature supporting the results obtained by the authors and add some conclusion to try explaining the outcomes obtained throughout the experiment.
Line 378-380: please add appropriate references for this statement.
In the main text are present numerous errors of grammar and syntaxis which need to be addressed. I would suggest the revision of the entire text by a native english speaker.
Author Response
Point 1: Line 69: Use the abbreviation PVL only after writing the word in full. Please make sure to check this issue throughout all the text.
Response 1: Thank you for your kind suggestion. Sorry for the abbreviate errors in the manuscript. We have changed the PVL to “Panton-valentine leukocidin (PVL)”, and changed the LPS to “lipopolysaccharide(LPS)”. Please see L86-87. We have also carefully checked this issue throughout all the text.
Point 2: Line 97: Use only BMECs, you already have written the word in full previously in the text. Please make sure to check this kind of issue throughout all the text.
Response 2: Thank you for your kind suggestion. We have changed the bovine mammary epithelial cells (BMECs) to “BMECs”. Please see L116, L1223. We have also carefully checked this kind of issue throughout all the text.
Point 3: Line 233-235: Please rephrase the sentence, it's not clear what you are trying to say.
Response 3: Thank you for your critical comments. According to your suggestion, We have changed the sentence to “Unlike the infection of yeast phase of C. krusei, which induced more cell apoptosis than that of the hyphae phase, the infection of hyphae phase of C. krusei caused more necrotic cells than the yeast phase”, Please check the revised manuscript. Please see L258-260.
Point 4: Please specify what stands for YC in figure 4, 6, 7 and 9 since this additional control in never mentioned in the main text.
Response 4: Thank you for your thoughtful comments on our work. According to your suggestion, we have added the relevant content description “BMECs were pretreatment with inhibition designated as the YC group, respectively”. Please check the revised manuscript. Please see L206-207.
Point 5: The authors should improve the discussion section with literature supporting the results obtained by the authors and add some conclusion to try explaining the outcomes obtained throughout the experiment.
Response 5: Thank you for your critical comments. According to your suggestion, We have added the relevant discussion content and references in the discussion section. And we also added the some conclusions. Please check the revised manuscript. Please see L432-449, L483-484, L468, L470, L487-491, L500-503.
Point 6: Line 378-380: please add appropriate references for this statement.
Response 6: Thank you for your kind reminder. We have added the relevant references in the discussion section, numbered [14], [29]-[31] in the revised manuscript. Please see L407.
Point 7: In the main text are present numerous errors of grammar and syntaxis which need to be addressed. I would suggest the revision of the entire text by a native english speaker.
Response 7: Thank you for offering the opportunity for revising the manuscript. We have carefully checked the grammatical and syntactic errors in the main text and corrected it. Please check the revised manuscript.

Reviewer 2 Report
The manuscript submitted for review with the title "Yeast and hyphae phases of Candida krusei induce apoptosis of bovine mammary epithelial cells via distinct signaling pathways" does not fall into the Subject Areas of the journal ( https://www.mdpi.com/journal/animals/about ), therefore I recommend that the manuscript be submitted to another journal specialized for similar type of studies – cellular, microbiological or veterinary medicine with different Subject Areas.
However, I have some recommendations for authors to improve manuscript quality:
Introduction: The problem with this type of mastitis is known and has been worked on since the middle of the 20th century (1960), and there are many published studies, including those from the last 2 years (for example, Gaviria and Montes, 2022), which are not mentioned by the authors. This indicates the importance of C. krusei in the pathology of mastitis in cows and other ruminants. The “Introduction” should be expanded to include data on fungal mastitis and specifically those caused by C. krusei worldwide and in China in general; add text on apoptosis (L385-390 of the "Discussion" section plus and others authors for example Lugli et al.2005) to have a transition to the aim of the research.
MM: The "MM" part is very well structured, but I have some recommendations and questions: MMs should be described in such a way that they can be replicated by other researchers. In this regard, the methodology, concentrations/amounts of inoculums for culturing the two phases of C. krusei should be specified. MAC-T cell line cultured roller or stationary? What are the cells cultured in - multiwell plates for cell culture (size), tubes or flasks for cell culture (as indicated on page 4 L152)? Is there a difference in the concentration of FBS in the growth and maintenance media? Page 4 L144 - use the abbreviation “PBS” because it is the abbreviation described and used above in the text. Some abbreviations are not described.
Results: The “Results” part is not written according to the results criteria - there is MM and discussion, but in this case this is necessary for a better understanding of the results obtained. However, the authors have to move the text containing the quotations/references and describe the result in another way (for example: subtitle 3.3, L271-280, 291-295, 314-315, 348-350). I have some technical recommendations: the subtitles should be in Italic ( same for MM); to bold the legend (Figures and Tables). Supplemental Fig.1 can be moved into the text as the main figure in it and have to be described.
In the description of the figures, describe the abbreviations J4, J8, S4 and S8 specified in MM. In this regard, I recommend replacing these abbreviations with the first letters of the respective phases ('y' and "h") to make the figures easier to analyze.
Discussion: The following text can be moved to "Introduction" - L386-390.
Why are apoptotic cells more in yeast than in hyphae (Fig.2A) and vice versa for necrotic cells, which are more in hyphae (Fig.2B)? or why “the hyphae phase is more likely to cause necrosis while inducing apoptosis of BMECs”? This should be discussed. The authors have to discuss the observed apoptosis and cell cycle of the host cells, the pathogenesis and the development of the mastitis
The authors have to discuss the reasons for the different results in the two phases of C. krusei development, thereby deepening and expanding the discussion. In this way, the literature used will increase, which will enrich the manuscript.
According to the changes of the "Discussion", the "Conclusions" should also be changed.
References: it is not as required by the Journal.
Author Response
Point 1: Introduction: The problem with this type of mastitis is known and has been worked on since the middle of the 20th century (1960), and there are many published studies, including those from the last 2 years (for example, Gaviria and Montes, 2022), which are not mentioned by the authors. This indicates the importance of C. krusei in the pathology of mastitis in cows and other ruminants. The “Introduction” should be expanded to include data on fungal mastitis and specifically those caused by C. krusei worldwide and in China in general; add text on apoptosis (L385-390 of the "Discussion" section plus and others authors for example Lugli et al.2005) to have a transition to the aim of the research.
Response 1: Thank you for your thoughtful comments on our work. We have added the relevant references in the Introduction section, numbered [3] in the revised manuscript. Please see L44. We also have added the relevant data on fungal mastitis in the Introduction section. Please see L56-L68.
Point 2: MM: The "MM" part is very well structured, but I have some recommendations and questions: MMs should be described in such a way that they can be replicated by other researchers. In this regard, the methodology, concentrations/amounts of inoculums for culturing the two phases of C. krusei should be specified. MAC-T cell line cultured roller or stationary? What are the cells cultured in - multiwell plates for cell culture (size), tubes or flasks for cell culture (as indicated on page 4 L152)? Is there a difference in the concentration of FBS in the growth and maintenance media? Page 4 L144 - use the abbreviation “PBS” because it is the abbreviation described and used above in the text. Some abbreviations are not described.
Response 2: Thank you for your kind suggestion.We have added the detailed data on culture in the methods section in the revised manuscript. Please see L110, L117, L170. Sorry for the abbreviate errors in the manuscript. We have changed the phosphate-buffered saline (PBS) to “PBS”, changed the PVL to “Panton-valentine leukocidin (PVL)”, and changed the LPS to “lipopolysaccharide(LPS)”. Please see L161, L86-L87. We have also carefully checked this issue throughout all the text.
Point 3: Results: The “Results” part is not written according to the results criteria - there is MM and discussion, but in this case this is necessary for a better understanding of the results obtained. However, the authors have to move the text containing the quotations/references and describe the result in another way (for example: subtitle 3.3, L271-280, 291-295, 314-315, 348-350). I have some technical recommendations: the subtitles should be in Italic ( same for MM); to bold the legend (Figures and Tables). Supplemental Fig.1 can be moved into the text as the main figure in it and have to be described.
In the description of the figures, describe the abbreviations J4, J8, S4 and S8 specified in MM. In this regard, I recommend replacing these abbreviations with the first letters of the respective phases ('y' and "h") to make the figures easier to analyze.
Response 3: Thank you for your critical comments. We have deleted the sentences related to the discussion in the results, and the sentences are deleted as follows:”Caspase-3/7 inhibitor I demonstrates significant and reversible inhibitor of caspase-3 and caspase-7, with a minimal effect on caspase-9, caspase-8, caspase-6, caspase-4, caspase-2 and caspase-1 [15]. While Z-IETD-FMK is a caspase-8 inhibitor by reducing the expression of CD25, but not the secretion of IL-2 or the production of IFN-γ. Z-LEHD-FMK is an irreversible inhibitor that specifically targets caspase-9, which does not impact the proliferation of normal cells [16]. Z-LEHD-FMK showed an ability to inhibit TRAIL-mediated apoptosis in HCT116 cell and 293 cell [17].”,”These data suggested distinct pathways mediated BMECs apoptosis in response to the infection of that the BMECs apoptosis induced by yeast phase and hyphae phase of C. krusei infections. BMECs apoptosis induced by yeast phase of C. krusei was mainly mediated by mitochondrial pathway, whereas the apoptosis induced by hyphae of C. krusei was mainly mediated death li-gand/receptor pathway.”,”These findings indicate that TLR2 plays an important role in ERK-mediated apoptosis in C. krusei yeast phase and hyphae phase-infected BMECs.”,”The results of the study indicated that both the yeast phase and hyphae phase of C. krusei demonstrated an ability to enhance the expression of ERK, p-ERK, JNK, and p-JNK.” Please check the revised manuscript.
According to your suggestion, We have changed all the titles, subtitles in the article to italics, and bolded the Figures. Please check the revised manuscript.
According to your suggestion, We have moved Supplemental Fig.1 into the article as Figure 2 and added the related result description. Please check the revised manuscript. Please see L241-252, L260-L266.
As for replacing J4, J8, S4 and S8 in the article with the initial letters y and h of yeast phase and mycelium phase, because the control group added with inhibitor is YC, we think it is better to use the original abbreviation in order to avoid misunderstanding.
Point 4: Discussion: The following text can be moved to "Introduction" - L386-390.
Why are apoptotic cells more in yeast than in hyphae (Fig.2A) and vice versa for necrotic cells, which are more in hyphae (Fig.2B)? or why “the hyphae phase is more likely to cause necrosis while inducing apoptosis of BMECs”? This should be discussed. The authors have to discuss the observed apoptosis and cell cycle of the host cells, the pathogenesis and the development of the mastitis
The authors have to discuss the reasons for the different results in the two phases of C. krusei development, thereby deepening and expanding the discussion. In this way, the literature used will increase, which will enrich the manuscript.
Response 4: Thank you for your critical comments. According to your suggestion, we have moved L386-390 text to Introduction section in the revised manuscript. Please see L82-85. According to your suggestion, We have added the relevant discussion content and references in the discussion section. Please check the revised manuscript. Please see L432-449, L468, L470, L483-484, L487-491.
Point 5: According to the changes of the "Discussion", the "Conclusions" should also be changed.
Response 5: Thank you for your critical comments. We have added the conclusion “while the hyphae phase of C. krusei caused more necrosis than the yeast phase. C. krusei yeast phase and hyphae phase could activate TLR2 in BMECs through the MYD88-dependent signaling, while hyphae phaseof C. krusei could activate TLR4 through the MyD88-independent signaling pathway in BMECs”. Please check the revised manuscript. Please see L500-L503.
Point 6: References: it is not as required by the Journal.
Response 6: Thank you for your comments. As for the format of the references, we will actively contact the editor of the magazine for modification. We have added the relevant references in the revised manuscript, numbered[3], [6]-[14], [14], [29]-[31],[39]-[47], [54],[55], [58]-[59].

Reviewer 3 Report
The purpose of this paper was to study whether the infection of this pathogen could induce apoptosis in bovine mammary epithelial cells. This is interesting, but there are still some shortcomings that need to be revised.
1.From the results, inflammation is more obvious. Why does the author focus on apoptosis instead of inflammation?
2. Flow cytometry is recommended to detect apoptosis.
3.There are some WB results showing pictures of poor quality, it is recommended to re-provide more clarity.
1. Units should be unified. such as hours insted of h, 5 minutes instead of 5 min;
Author Response
Point 1: From the results, inflammation is more obvious. Why does the author focus on apoptosis instead of inflammation?
Response 1: Thank you for your critical comments. Inflammation is the main symptom of dairy mastitis. We detected and analyzed the relevant inflammatory signaling pathways and cytokines, and found that both yeast phase and hyphae phase of C. krusei can activate the NF-κB pathways. The expression levels of p-IKB,p-P65, MYD88, IRF3, IRF5 and TLR2 were significantly increased. Cytokines TNF-α, IL-1β, IL-6, IL-8, IL-18, serum albumin and transferrin are involved in C. krusei Inflammatory lesions induced by yeast phase and hyphae phase infection of BMECs and rat mammary gland. We plan to publish the relevant research results in another article.
Point 2: Flow cytometry is recommended to detect apoptosis.
Response 2: Thank you for your comments. We have detected apoptosis by flow cytometry in the manuscript. According to your suggestion, We have moved Supplemental Fig.1 into the article as Figure 2 and added the related result description. Please check the revised manuscript. Please see L239-274.
Point 3: There are some WB results showing pictures of poor quality, it is recommended to re-provide more clarity.
Response 3: Thank you for your thoughtful comments on our work. We are also very concerned about the poor quality of some WB results. However, because some protein antibodies are not specifically bovine, they are polyclonal antibodies. We've replicated these protein bands many times, and these are the clearest. We will do our best to improve the quality of protein bands in future experiments.
Point 4: Units should be unified. such as hours insted of h, 5 minutes instead of 5 min.
Response 4: Thank you for your kind suggestion. we have replaced the h to hours, min to minutes throughout the article. Please check the revised manuscript.

Round 2
Reviewer 2 Report
The authors have improved the presentation of the manuscript. There are some omissions for example: To explain the abbreviation BMEC in the main text because it is only mentioned in the “Simple Summary” part. Authors have to explain the abbreviation ROS. The "MM" part does not mention inoculum amounts, number of cells/concentration of C. krusei used to infect the cultures, how is the amount of C. krusei cells determined? The described methodology should be able to be repeated by other researchers, to confirm these results or to apply to other microorganisms.
Important: In general, the growth medium contains 10% FBS and the maintenance medium 5% (up to 2%). The authors state that there is no FBS in the supporting environment. This can lead to faster cell death in uninfected cells as well.
All this, together with the lack of a detailed description, raises doubts about the conduct of the experiments and the results obtained.
In connection with the above, I allow myself to give two examples (about Candida) of how the "MM" section should be described:
https://www.sciencedirect.com/science/article/pii/S0141813017324133
https://www.scielo.br/j/mioc/a/kT5PxynL4s4mBsPHSygTH3x/?lang=en&format=html
Author Response
Point 1: The authors have improved the presentation of the manuscript. There are some omissions for example: To explain the abbreviation BMECs in the main text because it is only mentioned in the “Simple Summary” part. Authors have to explain the abbreviation ROS.
Response 1: Thank you for your kind suggestion. We have added the relevant explanation of BMECs in the Introduction section and explanation of ROS in the "MM" section of the revised manuscript. Please see L76 and L162.
Point 2: The "MM" part does not mention inoculum amounts, number of cells/concentration of C. krusei used to infect the cultures, how is the amount of C. krusei cells determined? The described methodology should be able to be repeated by other researchers, to confirm these results or to apply to other microorganisms.
Important: In general, the growth medium contains 10% FBS and the maintenance medium 5% (up to 2%). The authors state that there is no FBS in the supporting environment. This can lead to faster cell death in uninfected cells as well.
All this, together with the lack of a detailed description, raises doubts about the conduct of the experiments and the results obtained.
In connection with the above, I allow myself to give two examples (about Candida) of how the "MM" section should be described:
https://www.sciencedirect.com/science/article/pii/S0141813017324133
https://www.scielo.br/j/mioc/a/kT5PxynL4s4mBsPHSygTH3x/?lang=en&format=html
Response 2: Thank you for your thoughtful comments on our work. According to your suggestion, We have added the infected data on concentration of BMECs and C. krusei in the methods section in the revised manuscript. Please see L124-128.
Regarding the question about the maintenance medium without FBS, one of the reasons is that the yeast phase of C. krusei is easily transformed into the hyphae phase in the medium containing FBS, which will affect the experimental results. Another reason is that BMECs have strong vitality as the first defence barrier of mammary tissue. Our laboratory has conducted a preliminary experiment and found that BMECs can be cultured normally in the medium without FBS without affecting their vitality. The relevant experimental data can be seen from the "Cell Viability Assay" part of "MM" in published articles:
https://doi.org/10.3168/jds.2019-17619

Reviewer 3 Report
The units of the full text are still inconsistent, and it is recommended to review them carefully.
The units of the full text are still inconsistent, and it is recommended to review them carefully
Author Response
Point 1: The units of the full text are still inconsistent, and it is recommended to review them carefully.
Response 1: Thank you for your kind suggestion. Sorry for the inconsistent of units in the manuscript. We have carefully checked this issue throughout all the text and corrected errors in the revision. Please check the revised manuscript. Please see L184, L248-258 and L429-432.
